# Efficient Removal of Heavy Metals from Contaminated Sunflower Straw by an Acid-Assisted Hydrothermal Process

**DOI:** 10.3390/ijerph20021311

**Published:** 2023-01-11

**Authors:** Huijuan Song, Jun Zhou, Shilong He, Qiao Ma, Liang Peng, Miaogen Yin, Hui Lin, Qingru Zeng

**Affiliations:** 1Department of Environmental Science & Engineering, Hunan Agricultural University, Changsha 410128, China; 2Department of Materials Science, Hunan Agricultural University, Changsha 410128, China

**Keywords:** heavy metal, hydrochar, Cd, HCl, sunflower, species

## Abstract

The removal of heavy metals is crucial to the utilization of contaminated biomass resources. In this study, we report an efficient process of hydrothermal conversion (HTC) of sunflower straw (*Helianthus annuus* L.) to remove heavy metals. The effect of different HTC temperatures and concentrations of HCl additives on heavy metal removal efficiency was investigated. The results revealed that increasing the temperature or concentration of HCl promoted the transfer of heavy metals from hydrochar to liquid products during HTC. The heavy metals removed to the liquid products included up to 99% of Zn and Cd, 94% of Cu, and 87% of Pb after hydrothermal conversion with a temperature of 200 °C and HCl 2%. The species of heavy metals in hydrochars converted from unstable to stable with an increase in temperature from 160 °C to 280 °C. The stable fractions of heavy metals in the acidic condition decreased as the acid concentration increased. This aligns well with the high transfer efficiency of heavy metals from the solid phase to the liquid phase under acidic conditions. The FTIR indicated that the carboxy and hydroxy groups decreased significantly as the temperature increased and the concentration of HCl increased, which promoted the degradation of sunflower straw. A scan electron microscope showed that the deepening of the destruction of the initial microstructure promotes the transfer of heavy metals from hydrochars to liquid phase products. This acid-assisted hydrothermal process is an efficient method to treat biomass containing heavy metals.

## 1. Introduction

Soil pollution with heavy metals has become an important environmental issue as a result of rapid urbanization and industrialization [1,2,3,4,5]. Due to their strong mobility, heavy metals can pass through plant roots and accumulate in various plant tissues [6,7]. Phytoremediation is a cost-effective and environment-friendly approach to growing crops in heavy metal-polluted soil. Sunflowers have been widely adopted for the phytoremediation of heavy metal-contaminated soils owing to their high biomass production and capacity for metal accumulation [8]. Heavy metals remain primarily in the cakes in the process of extracting edible oil from oilseeds after phytoremediation, which helps them reach Chinese standards (GB2716-2005) [9,10]. However, the disposal of contaminated biomass after phytoremediation remains a problem [11]. It is necessary to dispose of contaminated biomass effectively to avoid causing secondary pollution and posing a threat to human health [11,12].

Heavy metal-contaminated biomass is treated with one or more of the following: direct disposal, compaction, composting, pyrolysis, leaching, and incineration (including combustion and gasification) [13]. Considering the environmental threat of heavy metal-contaminated biomass, thermal treatment (pyrolysis and incineration) has been widely employed. Gong et al. reported that pyrolysis effectively stabilized Cd, Cr, Zn, Cu, and Pb in ramie residues by converting the acid-soluble fraction of metals into a residual form and decreasing TCLP-leachable metal contents [11]. Liu et al. reported that organic acid leaching was an efficient approach for the detoxification of metal-containing plant incineration ash [14].

Hydrothermal conversion (HTC) has emerged as an environmentally friendly technology to recycle waste biomass. Compared with other thermal treatment technologies, such as incineration and pyrolysis, HTC shows superior performance in separating heavy metals without further contamination [15,16]. Furthermore, it can take advantage of hyperaccumulator biomass resources by recycling metals, chemicals, and hydrochar. Approximately 90% of accumulated Zn during phytoremediation is released from *Sedum plumbizincicola* biomass during subcritical hydrothermal liquefaction (HTL) reactions at an optimized temperature of 220 °C, and the release risk is mitigated via the HTL reaction for hydrochar production [17]. Lee et al. recently reported that hydrothermal carbonization reduces the concentration of heavy metals in hydrochars when heavy metal-containing sunflower residues obtained from a phytoremediation site are hydrothermally carbonized at 160–260 °C [18].

In this work, the hydrothermal conversion of heavy metal-containing sunflowers obtained from a phytoremediation site was investigated. The research focused on the distribution of heavy metals (Pb, Zn, Cu, and Cd) between solid and liquid phase products during HTC; additionally, the effects of HTC temperature and HCl additive concentration were considered. Moreover, the chemical species of heavy metals in hydrochars were examined to assess the transfer of heavy metals between the solid phase and liquid phase.

## 2. Materials and Methods

### 2.1. Preparation of Sunflower Straw

Sunflower straw was obtained from a contaminated field in Huanggu within Zhuzhou (27°34′27.64″ N, 113°13′2.21″ E), Hunan Province, southern China. Sunflower straw from harvested plants was rinsed with deionized water and then dried in an oven at 105 °C until it reached a constant weight. Next, the dried plants were cut and crushed to a definite granularity by 80 mesh. The dried samples were put in a 4 °C environment.

### 2.2. Hydrothermal Conversion of Sunflower Straw

The hydrothermal conversion of sunflower straw was conducted in a high-pressure autoclave reactor (316 stainless steel) with a volume of 300 mL. In a typical experiment, the reactor was loaded with 10 g of sunflower straw and 100 mL of deionized water. Next, the sealed and tightened reactor was placed in an oven and heated to 160, 200, 240, and 280 °C for 4 h. The reactor was allowed to cool to room temperature naturally. The solid and liquid phase products were separated by filtration. The liquid products were stored in a 4 °C refrigerator for further chemical analyses, and the obtained hydrochar was washed with distilled water and dried at 105 °C for 12 h. After that, the obtained hydrochars were placed in zip-lock bags for further analyses and labeled as HTC160, HTC200, HTC240, and HTC280.

The modified hydrochar was prepared with different concentrations of HCl. First, 10 g dry sunflower straw and 100 mL HCl solution with concentrations of 0.25%, 0.5%, 1%, and 2% (quality ratio) were poured into the reactor, and then the reactor was heated to 200 °C for 4 h. The post-processing process of the resulting product was similar to the above description. Depending on the addition of different hydrochloric acid concentrations, the resulting hydrochar was marked as HTC200-0.25, HTC200-0.5, HTC200-1, or HTC200-2. Each step was performed in triplicate.

### 2.3. Analytical Methods

Sunflower straw and hydrochar were digested in mixed acid (HNO_3_/HClO_4_, *v*/*v* 85:15%). The content of heavy metals in the liquid was determined using an Optima 8300 ICP-OES (PerkinElmer, Boston, MS, USA). The pH of hydrochar was measured in a 1:20 sample to water with a pH meter (Seven Compact, Mettler Toledo, Greifensee, Switzerland) [19]. Each experimental test was detected three times, and the results are shown as mean values. All the reaction agents that were previously mentioned in the extraction procedures were guaranteed reagents.

A sequential extraction process proposed by the European Community Bureau of Reference (BCR) was applied to determine the speciation of Pb, Zn, Cd, and Cu in hydrochar, which was presented in a sequence. The first part of the sequence was the acid-soluble/exchangeable fraction (i.e., exchangeable metal and carbonate-associated fractions) (F1). The second part was a reducible fraction (i.e., a fraction associated with Fe and Mn oxides) (F2). The third was an oxidizable fraction (i.e., oxidizable fraction bound to organic matter) (F3). The fourth was a residual fraction (F4). The details of the extraction approaches were reported previously [20]. The extracts of the first, second, third, and fourth steps were retained in a polyethylene container at 4 °C. The total concentration of heavy metals in the hydrochar was analyzed after digestion on a platen heater with a mixture of HCl/HNO_3_ (85:15% *v*/*v*).

### 2.4. Hydrochar Yield Calculation and Characterization

The yield of hydrochar (*Y_h_*) was calculated as follows [21]:$$Y_{h}=\frac{M_{h}}{M_{b}}\times 100\%$$
where *M_h_* and *M_b_* are the mass of hydrochar and biomass, respectively. A Fourier-transform infrared spectrometer (FT-IR) was utilized to characterize the carbon structure and functional groups of the samples using an 8400 Infrared Spectrometer (Shimadzu, Japan). The SEM images of hydrochars were obtained from a field-emission scanning electron microscope (SU-8010).

## 3. Results

### 3.1. The Hydrochar Yield and pH

The influence of temperature and HCl concentration on the yield and pH of hydrochar generated from sunflower stems is shown in Figure 1. As the temperature increased, the yield of sunflower hydrochar dropped sharply from 64.25% at 160 °C to 39.58% at 280 °C. It can be inferred that there was low carbonization of sunflower straw at low temperatures, and the carbonization increased at high temperatures, which led to a decline in yield. The pH of the hydrochar increased from 5.19 to 6.32. This result is possibly caused by the lack of acid functional groups, like carboxyl and hydroxyl groups, during carbonation. As the concentration of hydrochloric acid increased, the yield of sunflower hydrochar dropped from 59.96% (HTC200) to 31.90% (HTC200-2), and the pH dropped from 5.56 (HTC200) to 2.88 (HTC200-2).

### 3.2. The Content of Heavy Metals in Hydrochar

The influence of hydrothermal temperature on the concentration of heavy metals in hydrochar is shown in Table 1. The concentrations of heavy metals in sunflower stems were 5.98 mg kg^−1^ Pb, 487.48 mg kg^−1^ Zn, 14.59 mg kg^−1^ Cu, and 4.48 mg kg^−1^ Cd. The most abundant heavy metal in hydrochar was Zn, which was from 146.95 to 364.8 mg kg^−1^, and the second was Cu, which was from 7.79 to 23.63 mg kg^−1^. The concentration of all metals in hydrochar was higher than in straw except for Zn at 160 °C. As the temperature increased, the concentration of heavy metals in hydrochar decreased significantly. When the reaction temperature was 280 °C, the content of heavy metals in hydrothermal carbon was the lowest, and the total amount of Pb, Zn, Cu, and Cd decreased by 31.77%, 69.86%, 46.61%, and 59.82%, respectively. High temperatures can thus decrease the content of heavy metals and promote heavy metal transfer from the solid phase to the liquid phase. 

The content of heavy metals in hydrochar with different concentrations of HCl is shown in Table 1. The content of heavy metals in hydrochar decreased as the concentration of HCl increased. Compared with no HCl added, the concentration of heavy metals decreased. The concentration of Pb, Zn, Cu, and Cd had a minimum value when the concentration of HCl was 2%—they are 2.23, 19.4, 2.63, and 0.13 mg kg^−1^, respectively. This showed that the addition of HCl in hydrothermal carbonization is favored for the removal of heavy metals.

### 3.3. The Content of Heavy Metals in Liquid Phase Products

The influence of hydrothermal process temperature on the concentration of heavy metals in liquid phase products is shown in Table 2. The concentration of heavy metals in liquid products formed by HTC was low at 160 °C; however, it increased at 200 °C and from 240 °C to 280 °C, which was the last temperature at which the content of heavy metals in liquid products reached the maximum. This showed that high temperatures favored the removal of heavy metals from solid phase to liquid phase products via the hydrothermal procedure.

The influence of HCl addition on heavy metal concentration in the liquid product under the hydrothermal process is presented in Table 2. This shows that HCl had a positive impact on the content of heavy metals in liquid phase products. The concentration of heavy metals in the liquid phase increased as the concentration of additives increased. When the concentration of HCI was 2%, the Pb, Zn, Cu, and Cd had a maximum concentration of 0.47, 48.09, 1.35, and 0.44 mg·L^−1^, respectively. This indicates that the addition of HCI can promote the dissolution of heavy metals from hydrochar to liquid phase products. The higher the concentration of HCI, the greater the dissolution of heavy metals.

Pearson’s relation was tested between the HCl, Pb, Zn, Cu, and Cd concentrations in liquid. The Pb concentration significantly corresponded to the Zn (R = 0.995), Cu (R = 0.996), and Cd (R = 0.982) concentrations, with *p* values lower than 0.01. However, the HCl concentration showed no significant relation (*p* > 0.05) to these metal concentrations in liquid.

### 3.4. The Distribution of Heavy Metals between Hydrochar and Liquid Products during the Hydrothermal Process

The distribution of heavy metals between hydrochar and liquid products during HTC of sunflower stems at different temperatures is shown in Figure 2. As the temperature increased, the distribution of heavy metals in liquid products increased. The percentage reached 72%, 87%, 79%, and 84% of Pb, Zn, Cu, and Cd in the liquid phase at 280 °C, respectively. This suggested that temperature is important and affects the distribution of heavy metals during HTC of sunflower stems and that a high temperature is conducive to the transference of heavy metals from the solid to liquid phase.

The distribution of heavy metals between hydrochar and liquid products during HTC of sunflower stems with different concentrations of HCI is shown in Figure 3. The addition of HCl improved the migration of heavy metals from the solid to liquid phase. When the concentration of HCl was 0.25, the percentage of Zn and Cd in the liquid phase was as high as 94% and 90%, respectively. As the concentration of HCI increased to 1, only 2% of both Zn and Cd was recovered in the solid phase, and the content of Cu reached 93% in the liquid. The transference of Pb to the liquid phase was less than the other three metals. Compared with the no HCl addition treatment, the Pb, Zn, Cu, and Cd in liquid phase products increased 358%, 46%, 129%, and 46%, respectively. This suggested that the addition of HCl could promote the migration of heavy metals in hydrochar to liquid products. The higher the concentration of HCI, the more heavy metal transfers to the liquid phase.

### 3.5. The Species of Heavy Metals in Hydrochar

The relative percentages (% of total) of heavy metals existing in F1, F2, F3, and F4 in hydrochar with different temperatures are illustrated in Figure 4. The bioavailable fractions (F1 + F2) of Pb, Zn, Cu, and Cd were 52.96%, 64.24%, 32.67%, and 65.31% at 160 °C, respectively. These fractions dropped to 39.16%, 16.44%, 14.38%, and 17.29%, respectively, as the temperature increased to 280 °C. This indicated that the species of heavy metals converted from unstable to stable due to the increase in temperature.

The effects of HCl addition on the speciation of Pb, Zn, Cu, and Cd are illustrated in Figure 5. The bioavailable fractions (F1 + F2) of Pb, Zn, Cu, and Cd are all higher in the HCl medium than the pure water. The fractions (F1 + F2) of Pb, Zn, and Cu increased alongside an increasing acid concentration. Additionally, the bioavailable fraction of Cd remained at a high level at lower concentrations of HCl addition. 

### 3.6. The Surface Structure Characteristics of Hydrochars

To identify the functional groups on the surface of the prepared hydrochar, a FTIR analysis was carried out, and the results of hydrochar prepared at different temperatures with HCl addition are shown in Figure 6. The appearance of many peaks corresponded to functional groups on the surfaces of hydrochar. The single bonds of 3000–3700, 2800–3000, 1500–1650, 1650–1800, and 1400–1000 cm ^−1^ corresponded to the stretching vibration of -OH, C-H, C=C, C=O, and C-O, respectively [22,23,24]. Compared with raw material (RW), the characteristic peaks matched with -OH and C=O greatly weakened or even disappeared on the surface of hydrochar as the temperature increased, which indicated that the carboxyl and hydroxy groups decreased significantly, which then suggested that dehydration and decarboxylation had occurred [25]. This corresponded with a change in the pH of the hydrochar as represented in Figure 1a. The infrared spectrum absorption peak is different from the intensity change at different HCl concentrations. As presented in Figure 6b, the single bond intensity of 1650–1800 cm ^−1^ assigned to aromatic C=C and C=O increased with a higher concentration of HCl, which implies that the acid medium benefited the formation of a more stable aromatic structure through dehydration [21]. Both the hydrothermal conversion temperature and the HCl concentration increased, and the bonds at 1000–1400 cm ^−1^ assigned to C–O and C–O–C significantly weakened in both number and intensity. This variation suggests that a high reaction in temperature and HCl concentration can promote the degradation of hemicellulose and cellulose.

SEM was utilized to observe the surface physical morphology of the hydrochar, and the photographs of hydrochar that were prepared under different conditions are shown in Figure 7. Compared with RW, as the temperature increased, the morphology of hydrochar shifted. The images of hydrochar converted at 160 °C retained the original structure of the sunflower straw, and some small holes were created at 200 °C. As the temperature increased to 240 °C, a small number of microspheres formed, and the surface of hydrochar gained at 280 °C was covered with carbon microspheres. Carbon microspheres were also present in hydrochar in the presence of acid. Moreover, the higher the concentration of HCl, the more carbon microspheres of HTC appeared. The microspheres are likely cellulose-converted carbon spheres in hydrothermal carbonization [26,27]. Changes in the surface morphology of hydrochar may be caused by different reactions of sunflower straw during HTC with different temperatures or HCl concentrations. The formation of nanospheres and microfiber fragmentation is due to the destruction of the initial microstructure. Generally, the formation of carbon microspheres goes through the following steps. The cellulose, semi-cellulose, or lignin chains are hydrolyzed, then dehydration and fragmentation occur, and the pristine microfibers of corn straw are destroyed. Polymerization and condensation of those microfiber fragments occur in the soluble phase, the aromatization of hydrochars increases, and then carbon spheres are formed [28]. An increase in temperature or HCl concentration can accelerate the aromatization of hydrochars and thus promote the formation of more microspheres.

## 4. Discussion

### 4.1. The Effect of Temperature and HCl Addition on Hydrochar Yield and pH

The hydrothermal process is an environmentally friendly technology that can utilize entire waste biomass resources [17,29]. It can transform biomass into high-carbon solid products (hydrochar) and some liquid by-products. In this study, the yield of hydrochar depended on the reaction temperature. Compared with the hydrothermal temperature of 160 °C, the yield of HTC200, HTC240, and HTC280 significantly reduced. This may be related to the dehydration, decarboxylation, polymerization, and aromatization of cellulose, hemicellulose, and lignin in the raw material, which thereby leads to the decomposition and transformation of organic matter [30]. When the reaction temperature is low (160 °C), the hydrolysis of small molecule organic matter occurs, such as for oligosaccharides, phenolic resin glycosides, and fatty acids [30,31]. When the reaction temperature is high (160–280 °C), the raw material undergoes dehydration, aromatization, and condensation [26]. It can be inferred that the higher the reaction temperature, the deeper the carbonization of sunflowers, which leads to a declining yield of hydrochar. 

The Fourier infrared spectrum (Figure 6) analysis also indicated that the peaks corresponding to -OH, C=O, and C-O weakened or disappeared when the reaction temperature increased. This shows that the carboxyl and hydroxyl groups reduced during the hydrothermal reaction, which indicates the degradation of cellulose and hemicellulose. With an HCl addition from 0 to 2% under the hydrothermal procedure, the yield of hydrochar decreased by 28.06% (Figure 1b). This may be due to the acidic condition that favors decarbonylation and the breakage of glycosidic bonds in cellulose, which weakens polymerization and ultimately results in the degradation of cellulose macromolecules. Hydrochar yield is strongly affected by temperature, but the decrease in the yield is more moderate with increased HCl concentration. Previous studies also reported that the solid yield was not significantly influenced by the presence of HCl and other acids during the hydrothermal conversion of cellulose [32,33].

### 4.2. The Effect of Temperature and HCl Addition on Heavy Metal Distribution and Species

The hydrothermal method is considered one of the most promising techniques for treating contaminated biomass. However, the heavy metals in hydrochar have severely limited their applications. Previous studies showed that changing the hydrothermal process parameters (temperature, time, additive, and solid-to-liquid ratio) may alter the content of heavy metals in solid-liquid two-phase products [17,30,34]. It has been widely reported that a high reaction temperature facilitates the volatilization of heavy metals from the solid product during the heat treatment of hyperaccumulator biomass due to thermodynamic properties [35,36]. In this experiment, the content of heavy metals in hydrochar decreased alongside an increase in temperature and varied among metals. As the reaction temperature increased from 160 °C to 280 °C, the concentrations of Pb, Zn, Cu, and Cd decreased by 31.77%, 69.86%, 46.61%, and 59.82%, respectively. This may be caused by the different migration capabilities of Pb, Zn, Cu, and Cd during the hydrothermal conversion, depending on the balance between release and adhesion [37,38]. Most heavy metals are distributed in liquid phase products under acidic conditions, perhaps because the presence of H^+^ could inhibit the formation of heavy metal-containing precipitates or the surface-adsorbed form that could adhere to hydrochar. This could promote the release of heavy metal into the liquid phase in the form of metal cations, which could be captured via cation exchanging resin, adsorbent, or precipitant [39].

In contrast, the species of heavy metals converted from unstable to stable with the increase in temperature. Numerous studies demonstrated that the acid-soluble/exchangeable and reducible fraction of heavy metals could transform into oxidizable and residual fractions through complexation during hydrothermal conversion [21,40,41,42,43]. A possible explanation for the stabilization of heavy metals during HTC is that the unstable adsorbed or dissolved heavy metals (F1 + F2) were entrapped by the organic matter with a complex structure to form stable phases and fixation in the hydrochars, and the higher the HTC temperature, the stronger the effect. After treatment with acid, the Cd of residue fraction (F4) almost diminished in the hydrothermal char, and the acidic-solution and exchanged fraction (F1) were primary. It was indicated that Cd bounded with resolvability organic substances, such as starch, cellulose and hemicellulose. However, the insoluble three-dimensional lignin contained less Cd. On the contrary, the content of Pb and Zn bounding with lignin was relatively high. In the process of hydrochar formation, insoluble three-dimensional lignin was directly dehydrated and decarboxylated to form hydrochar [44], so the content of the F4 fraction was relatively high. Strong acid can effectively promote the dissolution of some lignin and release heavy metals when the acid concentration is higher than 1% (HCl/biomass in quality ratio).

High temperatures can promote the formation of complex substances between organics and heavy metals, which thus increases the content of oxidizable fraction (F3) in hydrochar. However, acid can inhibit this reaction. It was noted that acid has a relatively low inhibition effect on the reaction between Cd, Cu, and organic matters. Therefore, as the acid concentration increased, the proportion of easily oxidized Cd and Cu fraction (F3) increased or remained unchanged. It was indicated that Cd and Cu were easy to form very stable chelates with organic substances which were more stable than Pb and Zn. Furthermore, Pb was closely bound with iron manganese oxides and was less affected by acid. Its F2 content increased with the increase in acid concentration. It might be attributed to the fact that Pb belongs to hard alkali [45,46].

In short, Cd was the most active heavy metal in biomass, which may be related to the Cd bounding with low molecular weight and easily soluble organic compounds. However, Pb was the most stable heavy metal and was difficult to dissolve into the liquid phase, even if a large amount of acid existed. Moreover, it had a strong binding ability with iron manganese oxides.

## 5. Conclusions

HTC with HCl was developed to remove heavy metals from sunflower straw (*Helianthus annuus* L.) growth from Cd–Pb polluted soil. The species of heavy metals in hydrochars became stable with an increase in temperature from 160 °C to 280 °C. The stable fractions of heavy metals in the acidic condition decreased as the acid concentration increased. The heavy metals removed from the products included up to 99% of Zn and Cd, 94% of Cu, and 87% of Pb after HTC at optimized conditions with a temperature of 200 °C and HCl 2%. This method can provide a way to produce clean hydrochar, which has a stable and regular spherical structure, and could be widely applied in adsorption, catalysis, batteries, and other fields. It also provides a new method for the effective recovery of valuable metals from sunflower straw that has a high concentration of heavy metals.

## Figures and Tables

**Figure 1 ijerph-20-01311-f001:**
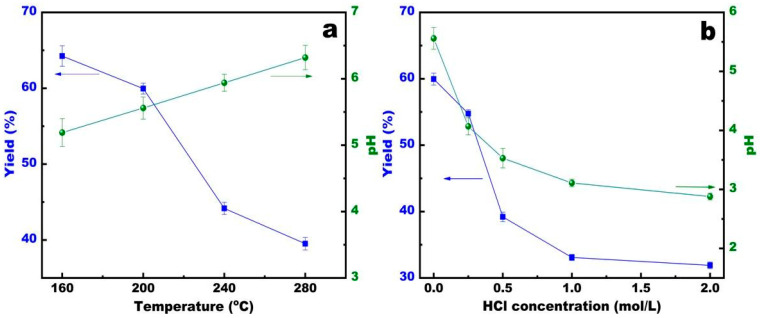
The yield and pH of hydrochar formed in different temperatures (**a**) and treated with different HCl concentrations (**b**).

**Figure 2 ijerph-20-01311-f002:**
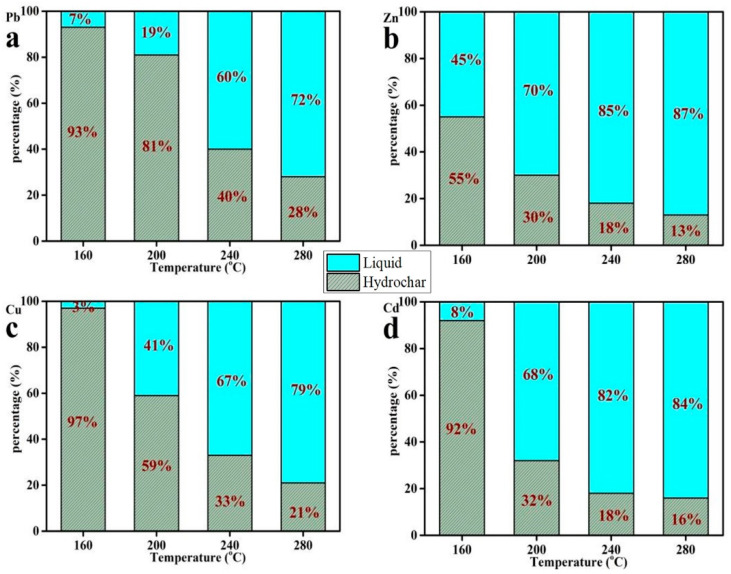
Different heavy metal fractions in liquid and hydrochar formed at different temperatures. Pb (**a**), Zn (**b**), Cu (**c**), Cd (**d**).

**Figure 3 ijerph-20-01311-f003:**
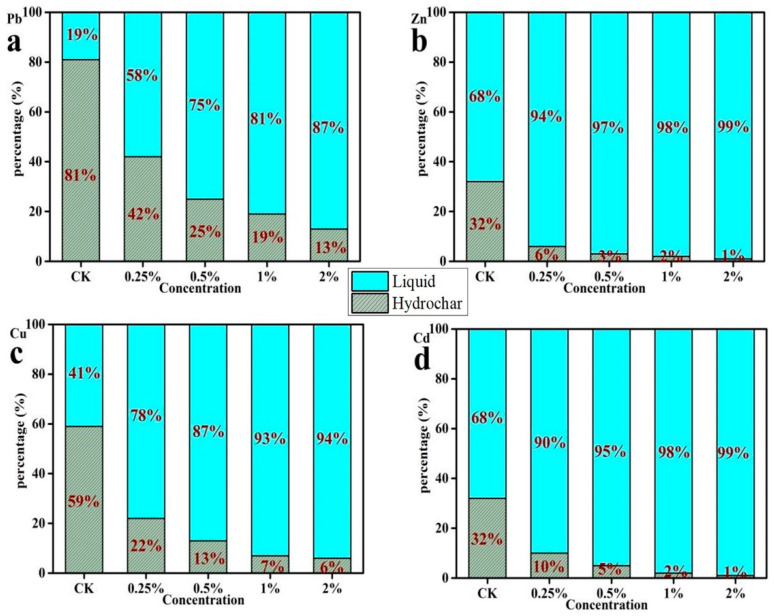
Heavy metal fractions in liquid and hydrochar formed in different HCl concentrations at 200 °C. Pb (**a**), Zn (**b**), Cu (**c**), Cd (**d**).

**Figure 4 ijerph-20-01311-f004:**
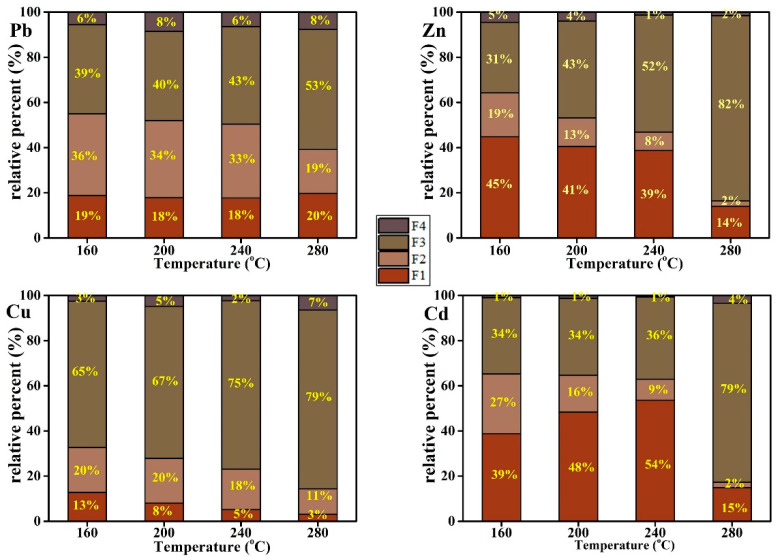
The relative percentages (% of total) of heavy metals (Pb), (Zn), (Cu), (Cd) existing in F1, F2, F3 and F4 in hydrochars with different temperature.

**Figure 5 ijerph-20-01311-f005:**
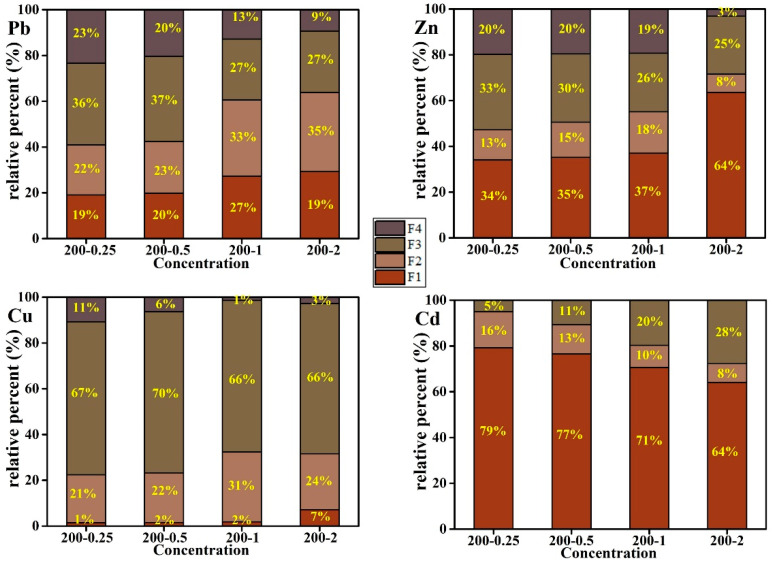
The relative percentages (% of total) of heavy metals (Pb), (Zn), (Cu), (Cd) existing in F1, F2, F3 and F4 in hydrochars with different HCl concentrations.

**Figure 6 ijerph-20-01311-f006:**
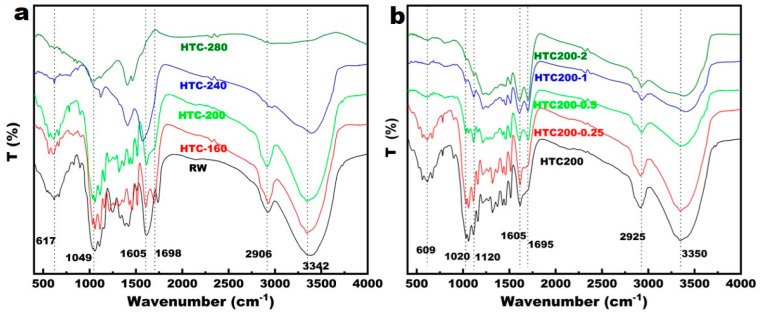
The FTIR of hydrochar prepared at different temperatures (**a**) and with different HCl concentrations (**b**).

**Figure 7 ijerph-20-01311-f007:**
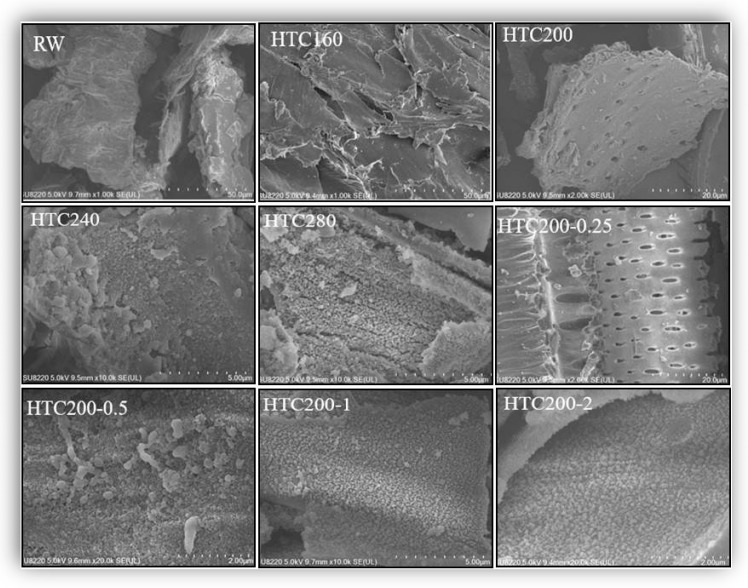
The SEM of hydrochar prepared from different conditions.

**Table 1 ijerph-20-01311-t001:** Concentration (mg·kg^−1^) of heavy metals in hydrochar formed by HTC at different HCl concentrations.

Sample	Concentration of Heavy Metals (mg·kg^−1^)			
	Pb	Zn	Cu	Cd
Untreated	5.98 ± 0.14 c	487.48 ± 39.59 a	14.59 ± 1.24 b	4.48 ± 0.29 b
HTC160	8.58 ± 0.38 a	364.80 ± 11.33 b	23.63 ± 1.54 a	6.25 ± 0.15 a
HTC200	7.30 ± 0.32 b	220.97 ± 18.24 c	13.84 ± 0.10 b	2.37 ± 0.14 c
HTC240	4.88 ± 0.06 d	182.32 ± 7.70 cd	10.90 ± 0.06 c	1.96 ± 0.06 d
HTC280	4.08 ± 0.12 e	146.95 ± 1.46 d	7.79 ± 0.11 d	1.80 ± 0.02 d
HTC200-0.25	4.13 ± 0.06 b	48.57 ± 1.64 b	5.27 ± 0.16 b	0.72 ± 0.03 b
HTC200-0.5	3.23 ± 0.08 c	31.27 ± 0.68 bc	4.57 ± 0.02 c	0.57 ± 0.02 c
HTC200-1	2.88 ± 0.14 c	27.32 ± 1.7 c	3.03 ± 0.16 d	0.24 ± 0.02 d
HTC200-2	2.23 ± 0.15 d	19.40 ± 0.23 c	2.63 ± 0.06 e	0.13 ± 0.01 d

Note: Different letters indicate significant differences (*p* < 0.05) from one another based on Duncan’s test results.

**Table 2 ijerph-20-01311-t002:** The concentration (mg·L^−1^) of heavy metals in liquid products formed by HTC at different conditions.

Samples	Metal Concentration (mg·L^−1^)			
	Pb	Zn	Cu	Cd
HTC160	0.04 ± 0.01 e	19.55 ± 0.58 f	0.04 ± 0.001 g	0.04 ± 0.004 g
HTC200	0.10 ± 0.01 e	30.27 ± 0.14 e	0.58 ± 0.03 f	0.30 ± 0.01 f
HTC240	0.32 ± 0.01 d	36.48 ± 0.91 d	0.96 ± 0.01 e	0.35 ± 0.01 e
HTC280	0.41 ± 0.01 b	39.91 ± 0.52 c	1.14 ± 0.01 dc	0.37 ± 0.003 d
HTC200-0.25	0.31 ± 0.01 d	41.65 ± 0.24 c	1.00 ± 0.02 dc	0.36 ± 0.004 d
HTC200-0.5	0.39 ± 0.01 c	45.48 ± 0.91 b	1.22 ± 0.05 b	0.39 ± 0.003 c
HTC200-1	0.42 ± 0.01 b	47.36 ± 0.18 a	1.30 ± 0.02 ab	0.42 ± 0.01 b
HTC200-2	0.47 ± 0.01 a	48.09 ± 0.17 a	1.35 ± 0.04 a	0.44 ± 0.01 a

Note: Different letters indicate significant differences (*p* < 0.05) from one another based on Duncan’s test results.

## Data Availability

The data that support the findings of this study are available from the corresponding author upon reasonable request.

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
