# Peer review of "Efficient Removal of Heavy Metals from Contaminated Sunflower Straw by an Acid-Assisted Hydrothermal Process"

_ijerph, 2023, doi:10.3390/ijerph20021311_

Round 1
Reviewer 1 Report
The heavy metal removal from straw growth from contamination farmland was an important issue in China. This MS discovered that the heavy metals can be removed from contaminated sunflower straw by acid-assisted hydrothermal process effectively, and the removal efficiency reached as high as 99% of Zn and Cd, 94% of Cu, 87% of Pb after hydrothermal conversion with the temperature of 200 oC and HCl is 2%. Specially, the Cd, with high toxicity, has been removed as much as 99%, indicated that it was an efficient technology to remove Cd. Therefore, this MS was very interesting and valuable. The minor review was needed before it published in the IJERPH.
1. The method of heavy metal concentration in liquid should be added.
2. The "℃" should be revised as the one form. And “hydrochar” and “hydrothermal carbon”, “HM” and “heavy metal” should be revised as one word overall the MS.
3. The language in this MS need to be improved.
Author Response
- The method of heavy metal concentration in liquid should be added.
R: The method of heavy metal concentration in liquid has been added. The content of heavy metals in liquid was determined using an Optima 8300 ICP-OES (PerkinElmer, USA).
- The "℃" should be revised as the one form. And “hydrochar” and “hydrothermal carbon”, “HM” and “heavy metal” should be revised as one word overall the MS.
R: The "℃" has been revised as the one form. And “hydrochar” and “hydrothermal carbon”, “HM” and “heavy metal” have been revised as one word overall the MS.
- The language in this MS need to be improved.
R: The language in this MS has been improved.
Reviewer 2 Report
This MS report an efficient process to remove heavy metals from sunflower straw by hydrothermal conversion, and analyzed the effect of heavy metal migration with different temperatures and concentrations of HCl. The experimental design is reasonable, the MS is interesting and has high reference value. However, there are some minor errors mas follows:
1.Line235-236,"……1500-1650, 1650-1800 cm-1……corresponded to the stretching vibration of ……C=C, C=O, respectively". While in Line 245-247, "1650-1800 cm-1 assigned to aromatic C-C and C-O increased ……, which implies …… more stable aromatic structure through dehydration". Such an argument is contradictory.
2.Take care to keep the font of the article consistent, such as "℃"
3. Line113,the abbreviation for sequential extraction process is BCR?
Author Response
- Line235-236,"……1500-1650, 1650-1800 cm-1……corresponded to the stretching vibration of ……C=C, C=O, respectively". While in Line 245-247, "1650-1800 cm-1 assigned to aromatic C-C and C-O increased ……, which implies …… more stable aromatic structure through dehydration". Such an argument is contradictory.
R: “C-C and C-O” have revised as “C=C and C=O”.
- Take care to keep the font of the article consistent, such as "℃"
R: This format has been reviewed.
- Line113,the abbreviation for sequential extraction process is BCR?
R: The European Community Bureau of Reference (BCR).
Reviewer 3 Report
The topic of this work and the multidisciplinary approach used are very interesting. However, in my opinion, the "results" section should be revised, removing the citations in the text which are more proper to the "discussion" section. Also, statistical tests could be applied to the dataset to investigate possible direct or inverse correlations among the variables and support your statements. The concluding paragraph can be improved.
Some other suggestions:
· Line 31 and 210: unnecessary capital letter “Heavy”
· In line 58: please remove the dot after “sedum” if it was not intentional.
· Please state more clearly the sentence on line 157. Also, did you perform any statistical analyses to better investigate the possible relationship among the variables?
· Line 166: “HEAVY METALS”
· Line 175: is the positive correlation you mentioned on this line supported by statistical analyses?
Author Response
- The topic of this work and the multidisciplinary approach used are very interesting. However, in my opinion, the "results" section should be revised, removing the citations in the text which are more proper to the "discussion" section. Also, statistical tests could be applied to the dataset to investigate possible direct or inverse correlations among the variables and support your statements. The concluding paragraph can be improved.
R:
- the "results" section have been revised and have removed the citations in the text to the "discussion" section.
The discussion content in result section has been deleted, as follow:
“This may be due to the enhanced decarboxylation which promotes the breakage of glycoside bonds in cellulose, resulting in reduced polymerization and degradation of cellulose macromolecules under acidic conditions [22].”
“Zhang et al. demonstrated a similar trend of Cd/Zn in a hyperaccumulator species (Sedum alfredii) hydrochar production [21]. Numerous studies demonstrated that the acid-soluble/exchangeable and reducible fraction of heavy metals could transform into oxidizable and residual fractions through complexation during hydrothermal conversion[23-26]. A possible explanation for the stabilization of heavy metals during HTC is that the unstable adsorbed or dissolved heavy metals (F1+F2) were entrapped by the organic matter with a complex structure to form stable phases and fixation in the hydrochars [27].”
“The bioavailable fraction of heavy metals increased with increased acid probably due to precipitation (as carbonates) and adsorption on hydrochars [28-29]. The bioavailable fraction of Cd remains at a high level at lower concentrations of HCl addition. This indicated that Cd is more active in response to the acid.”
These contents have been added in the discussion section.
- The statistical tests have been applied to the “3.3. The content of heavy metals in liquid phase products” to support the statements.
“The Pearson’s relation was tested between the HCl, Pb, Zn, Cu, and Cd concentration in liquid. The Pb concentration was significantly corresponding to the Zn (R=0.995), Cu (R=0.996) and Cd (R=0.982) concentration with p value lower than 0.01. However, the HCl concentration shown no significantly relation (p>0.05) to these metal concentration in liquid.”
- The concluding paragraph have been improved.
The conclusion section enhanced the detail results. Such as “The species of heavy metals in hydrochars converted to stable with an increase in temperature from 160 ℃ to 280 ℃. The stable fractions of heavy metals in the acidic condition decreased as the acid concentration increased.”
- The references have been revised correspondly.
- Line 31 and 210: unnecessary capital letter “Heavy”
R: The Heavy has been revised as heavy
- In line 58: please remove the dot after “sedum” if it was not intentional.
R: the dot after “sedum” have been removed.
- Please state more clearly the sentence on line 157. Also, did you perform any statistical analyses to better investigate the possible relationship among the variables?
R: It has been revised as “The concentration of Pb, Zn, Cu, and Cd had the minimum value when the concentration of HCl was 2%, they are 2.23, 19.4, 2.63, and 0.13 mg kg-1, respectively.
- Line 166: “HEAVY METALS”
R: It has been revised as “heavy metals”.
- Line 175: is the positive correlation you mentioned on this line supported by statistical analyses?
R: The positive correlation between the HCl concentration and heavy metal content in liquid was obtained from experimental results. However, the experimental data was too less to statistical analyses.
Round 2
Reviewer 3 Report
In my opinion, the authors have carefully considered the suggestions received and improved the article sufficiently.